Robustness of privacy-preserving collaborative recommenders against popularity bias problem

Gulsoy Mert 1 2
Yalcin Emre eyalcin@cumhuriyet.edu.tr 3
Bilge Alper 2
1 Distance Education Research Center, Alaaddin Keykubat University , Antalya , Turkey
2 Computer Engineering Department, Akdeniz University , Antalya , Turkey
3 Computer Engineering Department, Sivas Cumhuriyet University , Sivas , Turkey
Yang Jiachen
Electronic publication date: 2023 Jul 6
Publication date: 2023
Volume: 9
Electronic Location ID: e1438
Received 2023 Jan 30; Accepted 2023 May 24
Copyright: ©2023 Gulsoy et al.
Copyright year: 2023
Copyright holder: Gulsoy et al.
License: This is an open access article distributed under the terms of the Creative Commons Attribution License, which permits unrestricted use, distribution, reproduction and adaptation in any medium and for any purpose provided that it is properly attributed. For attribution, the original author(s), title, publication source (PeerJ Computer Science) and either DOI or URL of the article must be cited.
License URL: https://creativecommons.org/licenses/by/4.0/

Keywords: Recommender systems, Popularity bias, Privacy-preserving, Collaborative filtering, Unfairness

Funding: The Scientific and Technical Research Council of Turkey (TUBITAK) 122E040 This study is supported by the Scientific and Technical Research Council of Turkey (TUBITAK) under grant number 122E040. The funders had no role in study design, data collection and analysis, decision to publish, or preparation of the manuscript.

==============================
Recommender systems have become increasingly important in today’s digital age, but they are not without their challenges. One of the most significant challenges is that users are not always willing to share their preferences due to privacy concerns, yet they still require decent recommendations. Privacy-preserving collaborative recommenders remedy such concerns by letting users set their privacy preferences before submitting to the recommendation provider. Another recently discussed challenge is the problem of popularity bias, where the system tends to recommend popular items more often than less popular ones, limiting the diversity of recommendations and preventing users from discovering new and interesting items. In this article, we comprehensively analyze the randomized perturbation-based data disguising procedure of privacy-preserving collaborative recommender algorithms against the popularity bias problem. For this purpose, we construct user personas of varying privacy protection levels and scrutinize the performance of ten recommendation algorithms on these user personas regarding the accuracy and beyond-accuracy perspectives. We also investigate how well-known popularity-debiasing strategies combat the issue in privacy-preserving environments. In experiments, we employ three well-known real-world datasets. The key findings of our analysis reveal that privacy-sensitive users receive unbiased and fairer recommendations that are qualified in diversity, novelty, and catalogue coverage perspectives in exchange for tolerable sacrifice from accuracy. Also, prominent popularity-debiasing strategies fall considerably short as provided privacy level improves.

Introduction

Following the advent of online facilities through the Internet, many helpful digital platforms have been developed to support people in several aspects, such as daily activities (i.e., shopping, listening to music, watching movies) or interacting with people via social media. However, the amount of available information on these platforms has reached overwhelming levels in recent years, also known as the information overload problem, making it difficult for users to deal with (Jannach & Zanker, 2022). Recommender systems (RSs) are introduced as artificial intelligence-based solutions to overcome this problem by suggesting appropriate items/services to users’ expectations/desires (Ko et al., 2022). Since they provide many benefits for two fundamental stakeholders of the platforms, i.e., users and goods or service providers, many prominent digital applications for different domains, such as Netflix (https://netflix.com/), Twitter (https://twitter.com/), Spotify (https://spotify.com/), YouTube (https://youtube.com/), and Booking.com (https://booking.com/), have integrated such tools into their systems and continually attempted to improve them to provide better referrals to individuals.

The most effective approaches to providing individual recommendations are collaborative filtering (CF) techniques (Su & Khoshgoftaar, 2009), although some other scenario-specific practical approaches, like content- (Shambour et al., 2022) or knowledge-based filtering methods (Agarwal, Mishra & Kolekar, 2022) exist. By operating on a user-item preference matrix, the collaborative recommenders usually consider similarities between users or items and perform recommendation tasks by (i) computing a prediction score for items that are not experienced by users or (ii) generating ranked lists (i.e., top-N lists) including recommended items for users (Bobadilla et al., 2013). Over the last two decades, many different algorithms following this approach have been proposed, and also, there exists a rich literature with their main limitations (Koren, Rendle & Bell, 2022). Even if pioneering research has commonly aimed at handling their scalability, accuracy, or cold-start issues, their inclinations to feature a few popular items in their recommendations by disregarding many other ones, also referred to as the popularity-bias problem, have become one of the most critical concerns of RSs in recent years (Abdollahpouri et al., 2020; Elahi et al., 2021).

Such undesired bias propagation of the CF recommenders might be very problematic for users and service providers in real-world applications. On the one hand, users cannot receive qualified recommendations regarding beyond-accuracy aspects such as diversity, serendipity, and novelty, resulting in substantial decreases in their satisfaction with the platform (Yalcin & Bilge, 2022). On the other hand, items in the catalogue are not evenly represented in the recommendations, which leads to unfair competition in the market. Worse, if such weakness exists in the system, the platform might be subjected to manipulative attacks of malicious stakeholders aiming at featuring their products in recommendation lists or demoting rivals’ items to obtain financial benefits (Si & Li, 2020). Also, for specific application domains, such as social media, such weakness could be viciously abused to manipulate users by dragging them into ideological echo chambers (Cinelli et al., 2021). Because of these reasons, several recent studies in RSs have comprehensively evaluated the adverse effects of the popularity bias of the recommenders and developed various treatment strategies, also referred to as popularity-debiasing methods, to provide more balanced recommendation lists in terms of item popularity (Abdollahpouri, Burke & Mobasher, 2019; Yalcin & Bilge, 2021; Boratto, Fenu & Marras, 2021). Even if several popularity-debiasing methods exist, post-processing ones attempting to re-rank recommendation lists by penalizing popular items are the most utilized ones since they are independent of the internal mechanism of the considered recommendation approach and easily applied to the output of any recommenders.

The common agreement regarding why such a bias against popular items occurs in recommendations is the imbalances in the rating distribution observed in the original preference data on which the recommendation algorithm is trained (Abdollahpouri, Burke & Mobasher, 2018). More concretely, a few popular items typically receive most of the provided ratings in the dataset. In contrast, the remaining majority of items are evaluated by a small number of users, which ends up with user preferences dispersed with a long-tail shape across items, as presented in Fig. 1 for three benchmark data collections crawled for different application domains.

Another primary limitation of CF recommenders is that it might be challenging to collect confidential preference data of users because of some general reasons, such as the lack of functionality of user interfaces and the laziness of individuals. This issue also might be more problematic, especially for specific application domains like social media, since users avoid sharing their opinions on sensitive content due to data leakage concerns (Lu et al., 2022). Being unable to obtain genuine profiles of users mostly ends up with unqualified recommendations, especially regarding predictive accuracy, as traditional CF recommenders are trained on data consisting of unreliable ratings (Yargic & Bilge, 2019). Therefore, there has been considerable interest in privacy-preserving collaborative filtering (PPCF) to provide accurate predictions at a satisfactory level without violating such privacy concerns of individuals (Ozturk & Polat, 2015; Pramod, 2023). Although several PPCF schemes exist, the most prominent approach in central server-based recommenders is to apply randomized perturbation and obfuscation techniques during preference elicitation stages to mask (i) the list of actually rated items and (ii) the actual ratings given for these items (Polat & Du, 2003; Yargic & Bilge, 2019). Since the perturbation process is performed by adding random noise from a probability distribution with appropriate parameters, the aggregate still enables the central server to provide dependable yet private recommendations to users. Therefore, PPCF algorithms operate on a disguised version of the original preference collection where the perturbation level balances the confliction goals of accuracy and privacy.

Figure 1 The long-tail distribution of the provided user preferences in the (A) MovieLens-1M (movie), (B) Douban Book (book), and (C) Yelp (local business) data collections.

The random noise inserted into a user profile conceals the list of actually rated items by injecting fake votes into some unrated cells, which might lead to significant changes in the distribution of ratings across items. Therefore, this process might cause the level of observed popularity bias in the recommendations to be significantly altered, as the main reason for this issue is the imbalances in the dispersion of user preferences. Moreover, injecting fake ratings into user profiles alters each item’s total number of ratings received. Therefore, it would be expected that the performances of the popularity bias treatment methods, especially the post-processing ones, would significantly vary since they are mostly based on penalizing items based on their popularity rate (i.e., the total number of ratings received) while re-ranking recommendation lists (Abdollahpouri, Burke & Mobasher, 2019; Yalcin & Bilge, 2021). Although privacy-preserving recommendation systems are of crucial importance with the increasing focus on data protection regulations today, their popularity bias propagation behaviour remains elusive, and how well-known popularity-debiasing methods would perform on these systems is not investigated.

In this study, we attempt to present a comprehensive analysis evaluating how randomized perturbation-based privacy-preserving strategies affect popularity bias propagation in recommendations. In addition, we consider three prominent popularity-debiasing methods and scrutinize their performance with varying levels of privacy provided for users. For this purpose, we specify four user personas according to their desired privacy-protection level. The first persona, Utterly Unconcerned (UU), represents users who do not care about their privacy and require the highest possible level of accuracy in received recommendations. Note that UU also refers to users of traditional non-private CF recommenders. The second persona, i.e., Barely Bothered (BB) corresponds to users who are not so worried if their privacy is violated, and they pick low-level privacy parameters in a request of good accuracy. The third persona, Pretty Pragmatists (PP), represents users who do not compromise their privacy but are still interested in receiving decent recommendations. Therefore, they pick privacy parameters to balance accuracy and confidentiality. Finally, the fourth persona, Fiery Fundamentalists (FF), resembles highly concerned people with their privacy, so they are only willing to share extremely perturbed preferences sacrificing accuracy completely. In a nutshell, the main contributions of our study are given in the following.

• We present a comprehensive experimental analysis of how much the popularity bias is propagated in recommendations for four scenarios, i.e., UU, BB, PP, and FF, on three benchmark datasets using ten state-of-the-art recommendation algorithms of different families.

• We analyze how the adverse effects of the popularity bias problem on the accuracy and beyond-accuracy recommendation quality differentiate for such privacy level-based user personas.

• We consider three well-known popularity debiasing methods and observe whether they still perform similarly when BB, PP, or FF scenarios are considered rather than no privacy protection (i.e., the case of UU) provided.

The remaining sections of the study are organized as follows, respectively: (i) the ”Related work” section presents a review of both privacy-preserving methods for CF algorithms and popularity bias-concerned studies, (ii) the ”Constructing user personas based on varying privacy protection requirements” section explains how we construct user personas according to varying privacy protection requirements, (iii) the ”Experimental setup” section explains the experimental setups, including utilized datasets, recommendation algorithms, and evaluation protocols, followed by experimentation methodology and considered popularity-debiasing methods, (iv) the “Analyzing robustness of privacy-preserving collaborative recommenders against popularity bias issue” section analyzes how privacy-preserving strategies resist the popularity-bias problem, and the “Analyzing how popularity-debiasing approaches differently perform for privacy level-based user personas” section investigates how popularity-debiasing methods act for privacy-based user personas, and (v) the “Conclusion and future work” section concludes the study and introduces our future directions. Note that the source code accompanying this study is made publicly available (https://doi.org/10.5281/zenodo.7789914) for reproducibility purposes.

Related Work

Since our research is mainly based on two fundamental problems related to recommendation approaches, i.e., privacy protection and popularity bias issues, we organize this section into two sub-sections. The first provides a literature review on privacy-preserving strategies introduced for CF recommenders, while the second presents recent studies on the popularity bias problem and the treatment methods to cope with this issue.

Privacy-preserving approaches for CF recommenders

Recent years have seen a significant increase in research into the privacy concerns raised by conventional CF systems. Many studies have concentrated on developing PPCF approaches to address these issues (Wei, Tian & Shen, 2018). These approaches are commonly designed using k-anonymity, obfuscation, differential privacy, cryptography, and perturbation-based techniques.

For distributed recommendation systems, cryptographic techniques are frequently utilized to protect the privacy of individuals (Badsha, Yi & Khalil, 2016; Li et al., 2017). For example, an unsynchronized secure multi-party computation protocol was introduced by Li et al. (2016). This protocol incrementally computes similarities between items without requiring individuals to be present when the calculation stages are performed. As another example, Shmueli & Tassa (2017) have offered a distributed recommendation strategy in which vendors communicate encrypted data with a mediator over secure protocols to calculate predictions or ranking items in generating recommendation lists. Also, some recent studies have used homomorphic encryption to encrypt the quality of service values to recommend personalized, high-quality web services based on user and location data without revealing confidential information (Badsha et al., 2018). Furthermore, Li et al. (2017) have proposed a successful approach for the group recommender systems that aim to safeguard users’ privacy while making recommendations. The process of computing similarities between items is considered a probabilistic inference issue in Zou & Fekri (2015), and a semi-distributed belief propagation network-based approach is introduced for item-based PPCF systems. This way, disclosure of user preferences is prevented by including a subset of individuals in the network at a specific time.

Another crucial method for protecting user privacy is anonymization, which aims to break the link between particular users and their rating profiles. For example, a micro aggregation-based PPCF solution is suggested by Casino et al. (2015) to ensure user privacy protection by using k-anonymity masks. Chen & Huang (2012) explore achieving k-anonymity for large-scale and sparse database kinds (e.g., RSs) and introduce a k-anonymity heuristic approach based on clustering for protecting privacy in data sharing. In addition, Wei, Tian & Shen (2018) propose a (p, l, a)-diversification technique for improving the performance of existing k-anonymity methods, where p denotes previous information on the rating profile of the users and (l, a) denotes user diversity to increase the degree of provided privacy. As the last example of the anonymization-wised method, Zhang, Lee & Choo (2018) apply this strategy for a scenario of privacy-aware smart cities and model privacy issues in cities as a PPCF issue.

Differential privacy is another prominent successful mechanism employed in CF-based approaches to quantify the loss of privacy that may occur when a user’s personal preferences are considered when generating a suggestion. Dwork et al. (2006) is the pioneering study propounding the differential privacy concept, and it is later used by McSherry & Mironov (2009) for the RSs domain by randomizing actual user preferences using a private covariance matrix. A more recent study (Guerraoui et al., 2015) offers a distance-based differential privacy solution to alter real profiles by exchanging components at specific distances within the profile. Shen & Jin (2014) propose an alternative method that combines noise calibration, ensures theoretical privacy and utility, and allows for adjustable instance-based noise injection into the preference profiles of the users. Finally, Hou et al. (2018) suggest a two-stage methodology for neighbourhood-based differential privacy recommendations and private neighbour selection in the context of medical recommendations.

In CF systems, data modification techniques like data obfuscation and randomized perturbation are the most used to protect privacy in the literature. More specifically, the data obfuscation technique tries to conceal private information while maintaining access to the information required for making recommendations. For example, Parameswaran & Blough (2007) present a permutation-based data obfuscation technique for central server-based CF applications. Also, a user-based PPCF recommender utilizing semi-honest parties to execute additional calculations is proposed by Badsha et al. (2017). For distributed recommendation contexts, Boutet et al. (2016) suggest employing coarse-grained obfuscated versions of user profiles to reveal private information. Also, Elmisery & Botvich (2017) extend traditional sole obfuscation level-based methods to identify more reliable target users by considering arbitrary obfuscation degrees.

Randomized perturbation techniques (RPTs), an alternative to previous approaches, can safeguard user privacy by falsifying actual user preferences before they are sent to the recommendation server (Polat & Du, 2003; Polat & Du, 2005a; Polat & Du, 2005b). For example, Bilge & Polat (2013) develop a scalable PPCF technique based on bisecting k-means clustering and randomized perturbations. In another study (Gong, 2011), the privacy of user-profiles spread across various repositories in recommendation processes is protected by combining RPTs with secure multi-party computation techniques. By utilizing various levels and ranges of random values, Polatidis et al. (2017) suggest enhancing the RPT-based PPCF process. In addition, Liu et al. (2017) develop a hybrid approach to privacy preservation that combines RPTs with differential privacy to offer more robust user preference protection than current RPT-based approaches. Finally, Yargic & Bilge (2019) apply RPTs to protect individuals’ privacy in the multi-criteria recommender systems domain.

Apart from the RSs domain, some prominent privacy-protection mechanisms also exist, especially for machine learning, particularly federated machine learning (Yin, Zhu & Hu, 2021). For example, Xu et al. (2019) have introduced HybridAlpha to protect against privacy risks in federated learning using secure multiparty computation (SMC) protocol based on functional encryption. Another recent study offers a hybrid technique combining SMC and differential privacy to cope with the trade-off between accuracy and small data in parties, including small amounts of data (Truex et al., 2019). Also, Kerkouche et al. (2021) have introduced a privacy protection mechanism for federated learning in the context of medical research and patient care, relying on differential privacy to improve bandwidth efficiency by enabling confidentiality. Finally, Lu et al. (2022) introduce a federated learning approach to detect malfunctions that may occur in wind turbines in an organization in the energy sector. To ensure inter-institutional data privacy, they have enriched their proposed approach with a biometric authentication technique to reduce privacy-preserving risks.

As it is known, RSs typically need rating data of individuals to produce appropriate referrals to them. Therefore, the size/dimension of data collections for prediction purposes directly affects the recommendation quality; the need for big data also brings two different problems. One is the transfer, analysis, and processing costs of such massive data. Second, it is not easy to ensure the confidentiality of big data and protect user privacy. Some recent studies, therefore, have also been carried out on using distributed environments to protect data privacy (Omar et al., 2019; Bosri et al., 2020; Lin, Tian & Liu, 2021). For example, Bosri et al. (2020) introduce a privacy-preserving recommendation system, Private-Rec, by integrating artificial intelligence strategies with blockchain technology. This system provides a secure environment for users by utilizing distributed features where users’ data is employed with their permissions. Another recent study introduces a blockchain-based recommender system platform that prioritizes privacy preservation (Omar et al., 2019). The decentralized nature of blockchain allows clients to control their data and consent to its use in the analysis. This platform incentivizes clients to share their information by offering rewards such as points or discounts, which can be used for relevant recommendations provided by the online company. Finally, Lin, Tian & Liu (2021) develop a blockchain-based recommendation system using local sensitive hashing and differential privacy techniques in the blockchain environment. Thus, the cost of prediction calculations decreased while the quality of the recommendation and the level of user privacy increased.

Popularity bias issue of CF recommenders

One of the most critical topics in RSs is how fairly the system acts for two essential stakeholders, i.e., users and item/service providers, without propagating any bias in provided referrals (Abdollahpouri & Burke, 2019; Yalcin, 2022a). However, recent studies reveal that recommendation algorithms mostly fail to handle items in their produced ranked lists fairly due to several reasons, such as their internal mechanisms or characteristics of data where they are trained (Chen et al., 2020).

One well-known example of such an issue is the popularity-bias problem, which leads to recommendation lists dominated mostly by popular items, even if they would not be extremely desired (Abdollahpouri et al., 2020; Yalcin & Bilge, 2022). This problem, unfortunately, causes significant decreases in the beyond-accuracy quality of the recommendations, such as diversity, catalogue coverage, or novelty, and worse weakens the robustness of the system against some malicious attacks. Therefore, there has been a rising interest in scrutinizing its adverse effects on recommendations and developing practical popularity-debiasing methods to achieve more appropriate recommendations (Abdollahpouri, Burke & Mobasher, 2019; Yalcin & Bilge, 2021; Boratto, Fenu & Marras, 2021).

The initial research on this problem has mainly discovered its effects on recommendations for different application domains such as movies, music, travelling, and online courses. For example, Jannach et al. (2015) consider several CF recommenders with different categories, such as matrix factorization- and neighbourhood-based and observe that they intrinsically feature a small number of popular items repeatedly in their recommendations. They also analyze how finetuning parameters of such CF algorithms affect their bias toward popular items. Boratto, Fenu & Marras (2019) perform a similar analysis for massive open online courses by considering the same set of CF recommenders to analyze how they are biased towards the popularity of courses and their categories. Another pioneering study (Abdollahpouri et al., 2020) analyses how the effects of such an issue differ for users and item suppliers.

The literature also considers how the recommenders’ popularity bias unfairly affects individuals with different interest levels in item popularity. For example, in movie recommendations, Abdollahpouri et al. (2019) observe that this problem is more harmful to users interested in popular items in their original profiles when compared to selective ones who mostly rate niche ones. Such an analysis is later re-performed for music recommendations with a more extensive set of CF recommenders (Kowald, Schedl & Lex, 2020). Also, Yalcin & Bilge (2022) classify individuals based on the essential characteristics of their rating behaviour and observe that highly-interacting, selective, and hard-to-predict users receive extremely unfair recommendations, especially in terms of their original propensities on item popularity. Finally, Yalcin & Bilge (2023) have analyzed the effects of popularity bias on users with different personality traits in the big-five-factor model and explored that users avoiding new experiences and individuals who are less extroverted are subjected to more unfair recommendations regarding item popularity.

Researchers have been exploring ways to mitigate the issue of popularity bias in recommender systems. For this purpose, various methods have been developed that can be broadly classified into three categories: pre-processing, in-processing, and post-processing, depending on how they reduce bias in recommendations (Boratto, Fenu & Marras, 2021). Pre-processing methods typically aim to balance the distribution of user preferences by altering data used to train recommendation algorithms. One example is an approach that divides items into head and tail categories based on their popularity and uses different data sets for recommendations for each group (Park & Tuzhilin, 2008). Another pre-processing method introduced by Jannach et al. (2015) constructs user-item pairs where popular items are filtered out to achieve tuples where observed items are less-popular. After that, recommendation algorithms are trained on such constructed tuples. Additionally, a probability distribution function designed considering item popularity can give underappreciated items more prominence in recommended lists (Chen et al., 2018).

Several attempts have also been made to tackle the popularity bias problem in recommender systems, particularly through in-processing methods. These methods modify conventional recommendation algorithms to consider both relevance and popularity simultaneously; therefore, they are commonly considered algorithm-specific ungeneralizable solutions. Some prominent examples of these approaches include: (i) a method that estimates individual preferences for disfavored items and utilizes this information to penalize popular items during the recommendation process (Kamishima et al., 2014); (ii) an optimized variant of the RankALS algorithm that balances intra-list diversity and accuracy (Abdollahpouri, Burke & Mobasher, 2017); (iii) a personalized recommendation framework that estimates familiarity between items based on their popularity and prunes the most preferred ones in order to achieve a more balanced common-neighbor similarity index (Hou, Pan & Liu, 2018); (iv) a debiasing approach that minimizes the correlation between user-item relevance and item popularity, which helps to treat items in the long-tail equally (Boratto, Fenu & Marras, 2021); (v) and a recommendation method based on variational autoencoders that penalizes scores given to items based on their historical popularity to increase diversity in the recommendation results (Borges & Stefanidis, 2021).

On the other hand, post-processing methods for this issue can be applied to the output of any recommendation algorithm and involve re-ranking or creating a new list of recommendations that follows a specific constraint. One approach, proposed by Abdollahpouri, Burke & Mobasher (2018), involves calculating final scores by weighting the predicted ratings inversely proportional to the popularity of items and then using these scores to re-rank the recommended list. Similarly, Yalcin & Bilge (2021) propose two methods for popularity-debiasing in recommendations, multiplicative and augmentative, specifically tailored for groups of users rather than individuals. Both are based on a re-ranking strategy of items but differ in how they penalize items based on their popularity. More specifically, the multiplicative approach attempts to reduce the average popularity of the recommended items by using the item weights as a factor when computing the ranking scores. On the other hand, the augmentative approach takes prediction scores as the primary influence and adds the item weights as a factor to the ranking scores to achieve a reasonable level of accuracy in the final recommendations. Another renowned approach, presented by Abdollahpouri, Burke & Mobasher (2019), is the xQuad re-ranking approach, which balances the trade-off between ranking accuracy and the coverage of long-tail items.

As can be followed by the presented two sections, there is a rising interest in the RSs literature to protect individuals’ privacy and counteract the popularity bias of recommendation algorithms. However, to the best of our knowledge, there is no research analyzing how the bias propagation of CF recommenders differentiates when they are empowered with such privacy-protection mechanisms. In addition, it is required to perform more analyses of whether popularity-debiasing methods can still show similar performances when traditional CF recommenders are decorated with such privacy-preserving strategies.

Constructing user personas based on varying privacy protection requirements

Collecting accurate data from individuals is crucial for CF methods to provide qualified referrals. However, due to confidentiality issues, individuals usually avoid sharing their actual tastes for items/products, making it difficult to model them correctly, thus leading to poor referrals. Therefore, producing accurate recommendations by enabling users’ privacy has become one of the most important topics in RSs in recent years (Ozturk & Polat, 2015; Pramod, 2023).

PPCF approaches introduced towards such aim mainly attempt to conceal (i) the list of items truly rated by users and (ii) the genuine values of the ratings provided for them. The first can guard users against malicious attacks, such as unsolicited marketing, while the second can protect them from potential price or profiling discrimination attempts. As one of the most common and effective approaches to preserving users’ privacy, RPTs aim to randomly inject fake votes into some empty cells of the users’ profiles and randomly perturb each genuine rating in their profiles (Polat & Du, 2003; Yargic & Bilge, 2019). Thus, by obstructing the exposure of actual data of individuals without endangering the utility of preference data, these approaches are highly beneficial in providing privacy at a satisfactory level yet not sacrificing much from accuracy.

More concretely, these techniques suggest concealing a vote entry v by changing it with v + r, where r is a random value drawn from either a Gaussian or uniform distribution. The range of generated random values is adjusted by selected σu parameter, where σu = [0, σmax) and σmax is determined by the central server to regulate the distortion level of actual user preferences. Suppose that α is a constant and equals 3σu; the random values are drawn from either U−α,+α or N0,σu distributions. Moreover, RPTs inject random values as fake votes to user profiles’ arbitrarily selected empty cells. Here, the total number of empty cells to be injected is adjusted based on the selected βu parameter, where βu = [0, βmax) and βmax is determined by the central server relating to the density ratio of the entire user-item rating matrix.

In these settings, each user tunes their own σu and βu values based on desired privacy level and then masks their profiles, as described in Algorithm 1. Afterwards, they share their disguised profiles instead of the original ones with the service provider. Thus, a disguised preference matrix, i.e., Rm×n′, where m and n are the total numbers of users and items, respectively, is achieved on which the recommendation generation process will be performed via any proper CF recommender.

To comprehensively analyze how users with varying degrees of privacy are differently affected by the popularity bias issue in recommendations, we define four types of users according to their desired confidentiality protection level by applying Algorithm 1. The first, Utterly Unconcerned (UU), indicates individuals of the standard non-private CF approaches. These users are as completely reckless about their privacy and expect to receive recommendations with the highest possible accuracy. The second, Barely Bothered (BB), refers to users who prefer low-level privacy protection and do not get anxious if their privacy is slightly violated. The third, Pretty Pragmatists (PP), indicate individuals who pragmatically balance privacy and accuracy; therefore, they desire to receive decent recommendations as long as their privacy is not violated. Finally, the fourth, Fiery Fundamentalists (FF), refers to users who are hypersensitive about their privacy and never act as troublemakers regarding accuracy performance.

When guaranteeing the privacy of users by applying the RPTs, the selected values of σmax and βmax are crucial for them in adjusting their desired privacy level. Remember that higher (resp. lower) σmax and βmax values refer to more (resp. less) advanced privacy protection. Therefore, we construct these four user personas by considering the parameter tunings given in Table 1 when adopting the RPTs and examine each separately to observe the adverse effects for each privacy level-based user personas more clearly.

Table 1 Parameters of the constructed privacy-based user personas.

User Persona	σ max	β max	Dispersion	Privacy-level	
Utterly Unconcerned (UU)	0	0	N/A	No privacy	
Barely Bothered (BB)	2	5	Uniform/Gaussian	Low-level	
Pretty Pragmatists (PP)	3	10	Uniform/Gaussian	Mid-level	
Fiery Fundamentalists (FF)	4	25	Uniform/Gaussian	High-level	

Although provided privacy levels are intuitively observed in Table 1, we also present a methodology to quantify the concrete level of privacy obtained by βmax and σmax parameters. For this purpose, we perform separate sets of analyses for privacy quantification provided by these two parameters.

According to Algorithm 1, each disguised user profile contains genuine and fake ratings, filling βmax% of the empty cells. Therefore, privacy leakage by the server can only be realized by first estimating the exact set of genuine items with some uncertainty. Intuitively, such uncertainty can be quantified by estimating the probability of any randomly selected rating being genuine. Let a = {a1, a2, …, pa} denote the set of genuine ratings and f = {f1, f2, …, fe×βmax} define the set of inserted fake ratings. Then, the probability of an arbitrarily selected rating, r, to be genuine is Pr(r ∈ a) can be estimated as |a||a|+|f|. Note that such probability of being lower means a better level of provided privacy protection, and vice versa.

We present histograms of estimated probabilities for three personas in three datasets (i.e., MovieLens-1M (ML), Douban Book (DB), and Yelp) in Fig. 2. Note that the details of these datasets are also given in the following section. As expected, FF users mainly locate in the lowest probability bin in histograms, while PP and BB users tend to distribute into higher probability bins, indicating that personas obtain a declining level of confidentiality from FF to BB. Note that as the ratio of the number of ratings per user to the number of items in the dataset decreases in datasets, the effect of βmax on privacy level scales up, as can be observed in larger datasets such as Yelp and DB.

Figure 2 Histograms of estimated probabilities for three personas in the (A) MovieLens-1M (ML), (B) Douban Book (DB), and (C) Yelp datasets.

In the context of rating distribution, it is crucial to measure the privacy obtained by adding random noise to distinguish authentic votes from fake ones. Although central servers can differentiate between them, extracting genuine values from their perturbed z-score forms is necessary. To quantify the privacy of an additive noise-based perturbed variable, Agrawal & Aggarwal (2001) propose a metric based on differential entropy. This metric is applied in PPCF context by Polat & Du (2005b); Bilge & Polat (2013). In this context, the original user vector is represented by P, and the perturbing random data is represented by R, yielding U = P + R. The average conditional privacy of P is defined as Π(P|U) = 2H(P|U), where 2H(P|U) is the conditional differential entropy of P given U. Since P and R are independent random variables, the privacy level of P after disclosing U can be calculated as Π(P|U) = Π(P) × (1 − Pr(P|U)), where Pr(P|U) = 1 − 2H(U|P)−H(U). Assuming a normal distribution for P, the privacy levels, Π(P|U), for various perturbation levels are shown in Table 2. The output shows that the privacy levels increase with increasing perturbation of profiles, as expected.

Table 2 Privacy levels (Π(P|U)) for different personas.

Dataset	User Personas	
	BB	PP	FF	
ML	2.644	2.883	3.021	
DB	1.864	1.985	2.051	
Yelp	1.464	1.540	1.579	

Experimental Setup

In this section, we introduce the experimental settings, including utilized datasets, recommendation algorithms, and evaluation protocols, followed by experimentation methodology, and considered treatment methods for the popularity bias issue.

Datasets

In the experiments, we use three publicly available real-world benchmark datasets collected for different application domains, as in a very related recent study (Yalcin & Bilge, 2022); MovieLens-1M (https://grouplens.org/datasets/movielens/1m/) (ML) for the movies (Harper & Konstan, 2015), Douban Book (https://www.douban.com/) (DB) for the books (Shi et al., 2018), and Yelp (https://www.yelp.com/) for the local business reviews (Shi et al., 2018). User preferences in all utilized datasets are discrete and employ a five-star rating scale. Table 3 presents detailed information about the ML, DB, and Yelp datasets. Also, they have varying dimensions and sparsity ratios, as can be followed in Table 3, enabling us to analyze how such two important aspects of the datasets are related to potential unfairness issues for user personas with different privacy levels.

Utilized recommendation algorithms

It is very important to enrich the experiments with different parameters to analyze better the potential unfairness issues for individuals preferring varying privacy levels. The most convenient way to achieve it is to employ different algorithms in the recommendation generation phase. Indeed, this process can be described as generating recommendation lists using different algorithms after performing data perturbation and obfuscation operations to see how the privacy protection measures affect each algorithm’s bias towards popularity.

Therefore, ten different algorithms, two non-personalized and eight state-of-the-art approaches of four different CF families (i.e., matrix factorization-based, probabilistic, clustering-based, and neural networks-based), have been determined for recommendation purposes, as detailed in Table 4. Two non-personalized methods, i.e., ItemAverage (IA) and MostPopular (MP), are selected due to their wide usage in the literature to provide straightforward recommendations (Boratto, Fenu & Marras, 2019; Yalcin & Bilge, 2022). More specifically, the IA recommends items with the highest average rating, while the MP sorts items in descending order based on the number of their received ratings and then considers the most popular ones as recommendable. Therefore, by its nature, MP produces the same recommendation lists for each individual and denotes the maximum level of popularity bias that would be propagated. On the other hand, the remaining eight CF algorithms are selected as the best-performing ones from different families and applied via the Python-based library called Cornac (https://cornac.preferred.ai/) (Salah, Truong & Lauw, 2020). Also, for reproducibility purposes, we left hyperparameters of the algorithms set to their default values. Note that since the primary motivation of the present study is to evaluate whether recommendation algorithms propagate disparities among privacy level-based user personas, we do not provide detailed information about these algorithms’ internal mechanisms for clarity.

Table 3 Properties of MovieLens-1M (ML), Douban Book (DB), and Yelp datasets.

Dataset	# of users	# of items	# of ratings	Sparsity (%)	Rating average	# of Ratings per item	# of Ratings per user	
ML	6,040	3,952	1,000,209	95.8	3.58	253.1	165.6	
DB	13,024	22,347	792,062	99.7	4.05	35.4	60.8	
Yelp	16,239	14,284	198,397	99.9	3.77	13.9	12.2	

Table 4 Detailed information about utilized recommendation algorithms.

No	Family	Recommendation Algorithm	Abbreviation	
1	Non-personalized	Most Popular	MP	
2	Non-personalized	Item Average	IA	
3	Matrix factorization-based	Maximum Margin Matrix Factorization (Weimer, Karatzoglou & Smola, 2008)	MMMF	
4	Matrix factorization-based	Weighted Matrix Factorization (Hu, Koren & Volinsky, 2008)	WMF	
5	Matrix factorization-based	Hierarchical Poisson Factorization (Gopalan, Hofman & Blei, 2015)	HPF	
6	Probabilistic	Weighted Bayesian Personalized Ranking (Gantner et al., 2012)	WBPR	
7	Probabilistic	Indexable Bayesian Personalized Ranking (Le & Lauw, 2017)	IBPR	
8	Clustering-based	Spherical k-means (Salah, Rogovschi & Nadif, 2016)	SKM	
9	Neural networks-based	Neural Matrix Factorization (He et al., 2017)	NEUMF	
10	Neural networks-based	Variational Autoencoder for CF (Liang et al., 2018)	VAECF	

Evaluation metrics

Since the study aims to measure the effect of perturbation and obfuscation processes on popularity bias in different aspects, nine essential evaluation metrics are considered through the experiments. One of them, i.e., Average percentage of recommended items (APRI), is introduced by Yalcin (2022b) to measure the level of popularity of recommended items solely. The remaining metrics can be grouped under two main categories, i.e., accuracy and beyond-accuracy. More specifically, we employ the well-known normalized discounted cumulative gain (nDCG), precision, recall, and F1-score metrics for measuring accuracy performances, while average percentage of long-tail items (APLT), novelty, long-tail coverage (LTC), and Entropy for assessing beyond-accuracy recommendation quality. Here, although the selected beyond-accuracy metrics focus on measuring different aspects of the provided recommendations, such as popularity, diversity, coverage, novelty, and inequality, all of them indeed associate with the well-known fairness concept in RSs literature from the item perspective (Abdollahpouri & Burke, 2019). We describe these metrics below in detail, where i1, i2, …, iN denotes the items in the top- N list for user u, i.e., Nu, demonstrated by a recommendation algorithm.

Average percentage of recommended items (APRI): This metric calculates the average popularity of recommended items by the algorithms; therefore, higher APRI values for top- N lists indicate that the recommendation algorithm propagates a high-level of undesirable popularity bias (Yalcin, 2022b). More concretely, this metric first calculates the popularity ratio of each recommended item i, i.e., Pi, by considering the ratio of the total number of users who rate the corresponding item to the number of all users in the system. Then, it computes the average of the P values of the items in the Top- N list to achieve an APRIu score for the corresponding user u, as formulated in Eq. (1). (1) APRIu=∑i∈NuPiN

Precision, recall, and F1-score: To measure the accuracy quality of produced top- N recommendations, we employ well-known precision, recall, and F1-score metrics (Yalcin & Bilge, 2022). More specifically, the precision score of a top- N list (i.e., P@Nu) for a user u is calculated as the percentage of the recommended N items suitable for the user. On the other hand, the recall score of the corresponding recommendation list (i.e., R@Nu) is computed as the ratio of the number of suggested suitable items to the number of all rated items in u’s profile. Here, when determining whether a recommended item is suitable, we set the threshold value as 3.5, as the literature has reached a consensus that 4 and 5 are the affirmative ratings in a 5-star rating scale (Bobadilla et al., 2013). Finally, F1-score of the considered top- N list (i.e., F1@Nu) is calculated as the harmonic mean of its P@Nu and R@Nu scores, as formulated in Eq. (2); thus, it provides a balance between such two metrics measuring accuracy level in different aspects. (2) F1@Nu=2×P@Nu×R@NuP@Nu+R@Nu

Normalized discounted cumulative gain(nDCG): Another important accuracy metric used in the experiments is the nDCG, which considers the actual ratings of the recommended items and their positions in the produced top- N list (Yalcin & Bilge, 2022). Supposing that ru,i is u’s actual rating provided for item i, the DCGNuu and nDCGNuu values of the top- N recommendation list of u is calculated using Eqs. (3) and (4), respectively. (3) DCGNuu=ru,i1+ ∑n=2Nuru,inlog2n

(4) nDCGNuu=DCGNuuIDCGNuu

where IDCGNuu value is the maximum possible gain for u and is obtained by ideally ordering N items.

Average percentage of long-tail items (APLT): Another metric used in the experiments to evaluate the quality of ranking-based recommendations is the APLT, measuring what percentage of the recommended items are from the long-tail part of the popularity curve (Abdollahpouri, 2020). In other words, it demonstrates the ability of the recommendation algorithms to suggest niche items. When categorizing items as tail or head, the most commonly utilized approach is the Pareto Principle (Sanders, 1987), labelling items that receive 20% of the total ratings as head and the remaining ones in the catalogue as the tail items. Suppose that T is the set of assigned tail items via this principle, the APLT score of a top- N recommendation list produced for user u is calculated as given in Eq. (5). (5) APLTu=|i,i∈T∩Nu|N

Note that although APLTu reflects the diversification level belonging to a particular user. However, the overall APLT score is an average of such diversification levels, and the higher APLT scores do not necessarily mean that the considered algorithm can provide high-quality referrals in terms of diversity (Abdollahpouri et al., 2021). Nevertheless, lower APLT scores indicate a general lack of diversity in recommendation lists. Therefore, we also utilize Entropy as an alternative metric to measure the diversification level of all recommendation lists, which is explained in detail in the following.

Entropy: This metric mainly measures how often the recommendation algorithms suggest each item in the catalogue (Elahi et al., 2021). In other words, it reflects the inequality level across the frequency distribution of the suggested items. Therefore, higher entropy values for algorithms indicate more homogeneity in frequencies and diversity in recommendations, which is more desirable, especially for system administration, to enable fairer competition in the market. More concretely, the produced top- N list for each individual is merged by including the duplicated items. Suppose that such constructed union of each top- N list is represented with Nall and the set of all items in the catalogue {i1, i2, …, iK} is denoted with K, then the entropy value of a recommendation algorithm is computed using the formula given in Eq. (6). (6) Entropy=−∑i∈KPrilog2Pri

where Pr(i) denotes the relative frequency of item i appearing in Nall set.

Novelty: As a prominent beyond-accuracy metric, we also use the novelty in the experiments, which can be considered as the ability of the recommendation algorithms to present items that users did not taste before (Yalcin & Bilge, 2022). In other words, it measures what percentage of a top- N list produced for a user u has not been previously rated; thus, the capability of the algorithms to provide novel alternatives to users. Suppose that Iu is the set of items rated by user u, the novelty score of top- N recommendations provided for the corresponding user, i.e., Noveltyu, can be computed using Eq. (7). (7) Noveltyu=|i,i∈Nuandi∩Iu=0̸|N

Long-tail coverage (LTC): The last employed metric in the experiments is the LTC measuring how much the provided recommendations cover the tail part of the item catalogue (Abdollahpouri, 2020). We first construct a non-recurring aggregation of top- N lists produced for each user, which we refer to as ℕ. Contrary to the entropy calculation, the duplicated items in the individual top- N lists are eliminated here. Then, we determine an intersection item set, i.e., Iℕ∩T, by considering items that appear both in ℕ and T simultaneously, where the latter is the tail item set determined via Pareto Principle (Sanders, 1987). Finally, we calculate the ratio of the size of Iℕ∩T to the size of T, as in Eq. ??. Therefore, higher LTC values for algorithms indicate better coverage of the unpopular, i.e., tail, items in the recommendations. (8) LTC=|IN∩T||T|

Note that higher values of described metrics except for APRI indicate less-biased and more qualified recommendations in terms of accuracy and beyond-accuracy perspectives.

Experimentation methodology

In the experiments, we consider four scenarios described in Table 1, where users differ based on their desired privacy-protection level. Note that all experiments detailed in the following are realized for such four scenarios separately.

For each user profile scenario, we generate top-10 recommendations for users by using one of the previously explained algorithms. In doing so, we apply the well-known leave-one-out experimentation strategy (Kearns & Ron, 1997). Accordingly, we use a particular user as test data and the remaining ones as train data and compute predictions for all items for the user by running the considered algorithm on the train set. This process is performed repeatedly for each user in the dataset. Then, we select top-10 items for each user as the recommendation list based on their computed predictions in descending order.

To comprehensively evaluate how users concerned with varying privacy levels are differently affected by the recommendations, we compute precision, recall, F1-score, nDCG, APRI, and novelty value for the top-10 recommendation of each user and then calculate their averages to achieve final results. On the other hand, when computing the LTC and entropy performance of the algorithms, we consider the aggregation of all top-10 recommendations of users, as these metrics are not user-centric and operate on the set of all recommendations as a whole. Note that when reporting the experimental findings in the following section, we consider average values of 100 trials realized for each user-profile scenario to avoid potential inconsistencies during the randomization process of the perturbation and obfuscation phases.

The benchmark treatment methods for popularity bias problem

One of the main motivations of this study is to analyze how traditional popularity-debiasing methods’ performance varies when applied to disguised user profiles. To this end, we consider three benchmark post-processing treatment methods for the popularity bias issue concerning the recommendations’ different quality aspects. All methods aim to re-rank the recommendations by penalizing popular items and utilize a regularization parameter (i.e., λ) that weights predictive accuracy and one important beyond-accuracy perspective, such as diversity or coverage.

Two utilized popularity-debiasing methods are variants of the Enhanced Re-ranking Procedure proposed by Yalcin & Bilge (2021): Augmentative (Aug) and Multiplicative (Mul). Both approaches re-sort the items in the recommendation lists based on the synthetic ranking scores, which are estimated for items by inversely weighting their produced predictions and popularity ratios. However, they follow different mechanisms in producing such scores, i.e., Mul harshly penalizes popular items by weighting prediction scores inversely with the popularity ratio of items in a multiplicative way, and Aug uses prediction scores as a driving force by utilizing item popularities as minor factors. Thus, due to its design, the former can considerably improve beyond-accuracy quality, which might significantly decrease accuracy performance. On the other hand, the latter gives more importance to predictive accuracy while combating popularity bias issues as much as possible.

The third utilized algorithm is the famous xQuad (xQd) algorithm (Abdollahpouri, Burke & Mobasher, 2019), which focuses on achieving final recommendation lists for users by dynamically selecting items based on their categories: (i) head refers to the popular items, and (ii) tail refers to the unpopular (i.e., niche) items. The algorithm first selects a candidate item set by only considering the prediction scores produced by a recommendation algorithm. Then, it dynamically constructs the final recommendation lists based on the ranking scores of candidate items. Such scores are calculated at each iteration by weighting their prediction scores. The weighting process considers both the original propensities of users on the category of the corresponding item (i.e., head or tail) and the cover ratio of the current recommendation list for these categories.

Analyzing robustness of privacy-preserving collaborative recommenders against popularity bias issue

The first set of conducted experiments aims to observe how much the popularity bias of the recommenders affects recommendation quality for the base case, i.e., UU, and three scenarios of privacy protection for users, i.e., BB, PP, and FF. In other words, we measure how users with varying privacy concerns are differently exposed to the adverse effects of bias propagation of the algorithms. To this end, we perform several experiments by considering ten different recommendation algorithms (i.e., MP, IA, MMMF, WMF, HPF, IBPR, WBPR, SKM, NEUMF, and VAECF) and three datasets (i.e., ML, DB, and Yelp), which are explained above, and compute the obtained results for nine considered metrics for four privacy level-based scenarios. Accordingly, Figs. 3 and 9 present the obtained beyond-accuracy, i.e., APRI, APLT, entropy, novelty, and accuracy, i.e., precision, recall, F1-score and nDCG results for three datasets, respectively.

As can be seen in the y-axis of Fig. 3, regardless of the utilized algorithm and privacy level-based user personas, the highest APRI values are usually obtained for the ML, followed by DB and Yelp datasets, respectively. The main reason for this observation is that the average number of ratings for each item is higher in the ML compared to DB and Yelp datasets, as presented in Table 3. This fact inevitably leads to a higher popularity ratio for items and, thus, higher APRI values for the ML dataset. The MP algorithm is selected to demonstrate the maximum level of popularity bias propagation in recommendations. Therefore, as expected, the highest popularity bias in the recommendations is obtained with the MP algorithm for all settings. In addition, an unexpected finding is that the neural networks-inspired algorithms, i.e., NEUMF and VAECF, obtain almost identical APRI results with the MP algorithm, especially in the DB and Yelp datasets. Furthermore, the SKM can be considered one of the worst-performing algorithms in terms of APRI results for the ML dataset. Therefore, we can conclude that even if they are enhanced personalized CF algorithms, they, unfortunately, are biased towards popular items with extreme levels. On the other hand, the lowest APRI values are usually obtained with the IBPR or WMF among the utilized algorithms, especially in sparser and larger DB and Yelp datasets, which demonstrates that they are highly resistant to the popularity bias problem.

Figure 3 Obtained APRI results of the utilized algorithms for four different privacy level-based user profiles on (A) MovieLens-1M (ML), (B) Douban Book (DB), and (C) Yelp datasets.

The most important insight gained from Fig. 3, on the other hand, is that since the lowest APRI results are usually obtained for the FF case, followed by PP, BB, and UU, as the privacy level of the users increases, the average popularity of their received recommendations decreases. This finding is also valid for almost all cases, with minor exceptions. Observed differences in the obtained APRI results among privacy level-based user personas can also reach significant levels for some settings. For example, for the Yelp dataset, the UU is almost five times more than BB for the WBPR algorithm, or the PP is almost four times more than FF for the HPF algorithm. Therefore, we can conclude that privacy-protection mechanisms for CF algorithms not only eliminate the concerns of the users regarding their privacy but also improve the recommendation quality by alleviating the bias of the algorithms against item popularity. Moreover, such a finding is more concrete for DB and Yelp than the ML dataset; this is an important finding as data collections of real-world applications are typically sparser and larger.

BOX 1 Observation 1:

The most common privacy-preserving approaches for recommenders can efficiently eliminate users’ concerns regarding their confidentiality and provide notable decreases in the popularity ratio of recommended items simultaneously.

Contrary to the obtained APRI values, the achieved APLT results vary within the same interval for all utilized datasets, as can be seen in Fig. 4. The reason for this phenomenon is that it is a user-centric metric and calculates the average percentage of tail items in the produced top-10 recommendation lists. Since we have previously observed from achieved APRI results that the most vulnerable algorithms against popularity bias are MP, SKM, NEUMF, and VAECF, they are not talented in featuring tail (i.e., niche) items in their produced recommendation lists regardless of the considered privacy level-based user personas, as expected. On the other hand, the WMF and IBPR are the most prominent algorithms in providing diverse recommendations, as they achieve the highest APLT results. Also, the superiority of these algorithms is more visible for sparser and larger DB and Yelp datasets. The main reason for this observation could be that the size of constructed tail item set for the ML is notably smaller than those formed for these datasets; thus, the chance of including the tail items in the users’ recommendation lists would be significantly increased in these two datasets.

Figure 4 Obtained APLT results of the utilized algorithms for four different privacy level-based user profiles on (A) MovieLens-1M (ML), (B) Douban Book (DB), and (C) Yelp datasets.

Another crucial finding observed from Fig. 4 is that except for a few CF approaches (e.g., neural networks-inspired ones), the highest APLT scores of the recommendations are usually achieved for FF users, followed by PP, BB, and UU. This result reinforces that as the privacy level of users increases, their received recommendations become more qualified in terms of diversity aspect. Additionally, such differences in the obtained APLT scores are clearer for Yelp and DB datasets. The main reason for this consequence is that increasing privacy protection of users by including fake ratings in their profiles leads to significant changes in the distribution of the ratings, which makes algorithms more robust against popularity bias, as previously discussed in the APRI results. This, in turn, intuitively increases the probability of including tail items in the final recommendations.

Similar to the achieved APLT values, the obtained entropy, novelty, and LTC scores of the algorithms spread out in the same range, i.e., [0, 1], for all datasets regardless of the privacy level-based user profiles by their design, as can be followed by Figs. 5 and 7, respectively. We also observe from Fig. 5 that the lowest information entropy results are usually obtained with SKM, NEUMF, and VAECF algorithms. Therefore, we conclude that neural networks-inspired algorithms are highly unsuccessful in equally representing candidate items in the catalogue in their recommendations.

Figure 5 Obtained Entropy results of the utilized algorithms for four different privacy level-based user profiles on (A) MovieLens-1M (ML), (B) Douban Book (DB), and (C) Yelp datasets.

Compared to other algorithms, such weaknesses of the SKM, NEUMF, and VAECF also appear in providing recommendations that are qualified in terms of novelty and catalogue coverage, as seen in Figs. 6 and 7, as they usually have the lowest novelty and LTC scores for each considered privacy level-based user personas. Also, since their LTC results are generally around zero, we can conclude that these algorithms, including HPF, are highly inefficient for item/service providers attempting to sell each product at least once via suggestions covering the whole catalogue. Also, we observe that the novelty scores of all algorithms, including neural networks-based ones, are lesser for ML than other DB and Yelp datasets. The main reason for this result is that the average number of ratings per user is around 166 for ML, while 61 and 12 for DB and Yelp datasets, respectively, as can be followed by Table 3. Such differences in data collections make the recommendation algorithms superior to the DB and Yelp datasets in locating items unknown/inexperienced by the individuals in the recommendations, resulting in higher novelty scores for such sparser and larger datasets.

Figure 6 Obtained novelty results of the utilized algorithms for four different privacy level-based user profiles on (A) MovieLens-1M (ML), (B) Douban Book (DB), and (C) Yelp datasets.

Figure 7 The obtained LTC results of the utilized algorithms for four different privacy level-based user personas on (A) MovieLens-1M (ML), (B) Douban Book (DB), and (C) Yelp datasets.

On the other hand, the IBPR and WMF, especially the former one, achieve the highest entropy, novelty, and LTC outcomes regardless of the considered datasets and privacy level-based user personas, as seen in Figs. 5–7. In other words, the recommendations of these two algorithms are highly qualified in terms of covering and representing items in the whole catalogue fairly and featuring items that individuals have not experienced. Remember from the APLT analysis that IBPR and WMF are the most prominent algorithms in featuring items in the long-tail part of the catalogue in their recommendations, as well.

BOX 2 Observation 2:

In producing qualified recommendations in terms of beyond-accuracy aspects, i.e., diversity, novelty, equality, and catalogue coverage, the most successful algorithms are IBPR and WMF, while the least successful ones are mostly neural networks-inspired algorithms, i.e., NEUMF and VAECF.

Comparing the privacy level-based user personas, we observe from Fig. 5 that the highest entropy values are usually achieved for users highly concerned with their privacy, i.e., FF users, followed by PP and BB, respectively. Therefore, as the privacy-protection level increases for individuals, the likelihood of receiving more qualified recommendations where items are equally represented improves with minor exceptions. As can be seen in Fig. 6, this advantage for privacy-sensitive users is also valid for obtaining novel recommendations, as the highest novelty scores are again obtained for FF, followed by PP and BB, respectively.

However, the differences in the novelty scores among the privacy level-based user personas seem to be more visible for the ML while limited for sparser and larger DB and Yelp datasets; this occurs due to the same reason related to dataset characteristics explained above. Additionally, even if such differences among privacy level-based user personas are not quite visible for the LTC results achieved for DB and Yelp datasets, it can be concluded that the most sensitive users in the ML against privacy protection can receive more qualified referrals in terms of catalogue coverage.

BOX 3 Observation 3:

As the level of privacy provided for users improves, their referrals also become more qualified in terms of diversity, novelty, catalogue coverage, and fairness (i.e., equality in representing items).

Ranking accuracy is one of the most important criteria for assessing the quality of provided recommendations. Therefore, we perform an additional set of experiments where we comprehensively analyze how recommendation algorithms perform differently for privacy level-based user personas in terms of accuracy by utilizing four important accuracy-oriented metrics (i.e., precision, recall, F1-score, and nDCG). To this end, we present the obtained precision and recall scores of ten algorithms for four user profiles in Fig. 8, and F1-score and nDCG results in Fig. 9.

Figure 8 The obtained precision and recall outcomes of the utilized algorithms for four different privacy level-based user personas.

Precision results are presented in (A), (C), and (E) for the ML, DB, and Yelp datasets, respectively, while recall results are presented in (B), (D), and (F).

Figure 9 The obtained F1-score and nDCG outcomes of the utilized algorithms for four different privacy level-based user personas.

F1-score results are presented in (A), (C), and (E) for the MovieLense-1M (ML), Douban Book (DB), and Yelp datasets, respectively, while nDCG results are presented in (B), (D), and (F).

Due to the well-known trade-off between accuracy and beyond-accuracy recommendation quality, we expect that the algorithms performing well in beyond-accuracy aspects would not provide satisfactory ranking accuracy or vice versa. Such a contradiction is also valid for privacy level-based user personas, i.e., users cannot simultaneously receive recommendations of good quality in both beyond-accuracy and accuracy aspects. The accuracy-focused experimental results validate this phenomenon. For instance, the algorithms succeed in beyond-accuracy quality, e.g., the IBPR, has usually lower precision, recall, F1-score, and nDCG results than others, as can be seen in Figs. 8 and 9. On the other hand, the algorithms that are not succeeding in beyond-accuracy quality, e.g., neural networks-inspired ones (NEUMF and VAECF), show relatively better performance than others in providing accurate recommendations. Also, we can conclude that the WBPR is another successful algorithm to achieve accurate recommendations. However, the obtained results demonstrate that the WMF algorithm shines out among other algorithms in ranking accuracy when utilized without any privacy protection mechanism, i.e., the case of UU. This is an interesting finding since we have previously discussed that it is one of the best-performing algorithms regarding beyond-accuracy aspects. Therefore, we can conclude that the WMF can be a good option for a traditional recommendation scenario where no privacy protection is applied.

BOX 4 Observation 4:

For traditional recommendation scenarios where any level of privacy protection is provided, the WMF is the most prominent algorithm considering accuracy and beyond-accuracy aspects simultaneously. Also, the most accurate recommendations are usually obtained by the VAECF and WBPR algorithms.

As can be followed by Figs. 8 and 9, users receive more accurate recommendations in the UU case compared to privacy-preserving scenarios for almost all metrics; this is also valid for all settings except for non-personalized primitive MP and IA recommendation methods. When comparing privacy level-based user personas, the lower precision, recall, F1-score and nDCG results are obtained for FF, followed by PP and BB. This finding is valid for all personalized CF algorithms in the ML; however, the differences among such privacy level-based user personas seem to be limited for NEUMF and VAECF algorithms on DB and Yelp datasets. Therefore, we can conclude that as the privacy degree adjusted for users improves, the accuracy of their received recommendations becomes worse in general. The main reason for this consequence is that the algorithms can model individuals better when their original profiles are handled and thus provide highly accurate recommendations; however, their performances significantly diminish when their disguised profiles are considered. Note that this finding parallels previous research on PPCF topics (Bilge & Polat, 2013).

BOX 5 Observation 5:

As the level of privacy provided for users improves, the accuracy of their received recommendations diminishes.

Analyzing how popularity-debiasing approaches differently perform for privacy level-based user personas

This section presents the experimental results to investigate how the conventional popularity-debiasing methods perform for varying privacy level-based user personas. To this end, we consider three benchmark treatment methods previously explained, i.e., Aug, Mul, and xQd, and measure their effects on user profiles with varying privacy levels (i.e., BB, PP, and FF) and original profiles (i.e., UU). We realize this set of experiments on the Yelp dataset, as it is sparser and larger than others, like in most real-world applications. In addition, since the differences between privacy level-based user personas are more obvious for the HPF algorithm, we only consider this algorithm for the recommendation phase to see the debiasing methods’ effects better and for the sake of clarity. In these settings, we present the obtained improvements/deteriorations in beyond-accuracy (i.e., APRI, APLT, entropy, novelty, and LTC) and accuracy (i.e., precision, recall, F1-score, and nDCG) results with percentages when the Aug, Mul, or xQd applied in Figs. 10 and 11, respectively.

Figure 10 The obtained (A) APRI, (B) APLT, (C) entropy, (D) novelty, and (E) LTC improvement or deterioration level when the Aug, Mul, and xQd popularity-debiasing methods are applied for different privacy level-based user personas in the Yelp dataset.

Figure 11 The obtained (A) precision, (B) recall, (C) F1-score, and (D) nDCG improvement or deterioration level when the Aug, Mul, and xQd popularity-debiasing methods are applied for different privacy level-based user personas in the Yelp dataset.

As can be seen in Fig. 10, we observe that the most significant reductions for the APRI are obtained for FF, followed by PP, BB, and UU. Considering that lower values for this metric indicate less-biased recommendations against item popularity, we can conclude that more sensitive individuals to their confidentiality benefit more from such treatment methods against popularity bias. On the other hand, we have interestingly observed that the debiasing methods show worse performances for such privacy-sensitive users in terms of accuracy. Note that the most severe deteriorations in ranking accuracy are observed for FF, followed by PP, BB, and UU, as can be followed by all precision, recall, F1-score, and nDCG results given in Fig. 11. This observation is interesting since the accuracy-concerned metrics usually check how much the recommended items hit with the items rated in users’ original profiles. Therefore, we expect that the overall accuracy performance improves with recommending popular items.

Contrary to this pattern, on the other hand, the greatest improvements in APLT, entropy, novelty, and LTC results of the recommendations are obtained for UU, followed by BB, PP, and FF. This trend is also valid for all utilized popularity-debiasing methods, i.e., Aug, Mul, and xQd. In other words, the considered popularity-debiasing methods are more effective when the original profiles of the users are considered, and their ability to enhance beyond-accuracy recommendation quality decreases as the users’ privacy level increases. This is an unsurprising finding since we have previously observed that as the level of privacy provided for users increases, the beyond-accuracy qualities of their received recommendations are notably improved. Thus, the effects of the utilized debiasing methods become limited for individuals who are more concerned about their privacy. In addition, such privacy-sensitive individuals also seem to be disadvantageous in terms of accuracy, as the highest decrements in ranking accuracy are usually obtained for FF, followed by BB, PP, and UU, as discussed above.

One of the most important findings is that applying one of the highly prominent popularity-debiasing methods (i.e., the xQd) for FF users even significantly deteriorates recommendation quality (note that its improvements are negative for APLT and novelty while positive for APRI). This important finding demonstrates that applying privacy-protection procedures to user profiles might render some well-performing popularity-debiasing methods useless. The empirical outcomes also suggest that the most successful popularity-debiasing method differs for privacy level-based user personas depending on the considered quality aspect.

BOX 6 Observation 6:

As users’ privacy protection improves, the popularity-debiasing methods can perform better in achieving less-biased recommendations against item popularity. However, their effects on both accuracy and beyond-accuracy recommendation quality considerably diminish for users who are more sensitive to their confidentiality. Some of such treatment methods even become useless for fiery fundamentalists, i.e., users hypersensitive to their privacy.

Conclusion and Future Work

One main task of recommender systems is to produce appropriate referrals to users by protecting their privacy. The most common approach for this aim is masking the list of rated items and actual ratings provided for these items. Then, any collaborative recommender is trained on disguised preference data rather than the original user-item matrix. Since the main reason for the well-known popularity bias issue of the recommenders is the observed imbalances in the rating distribution, such privacy-protection procedure might lead to significant changes in the degree of observed bias in the recommendations and, worse, make treatment methods developed for this problem less useful.

Therefore, in this study, we mainly analyze how popularity bias inclinations of the recommenders vary for four user personas defined according to varying privacy protection levels. We also investigate how the performances of three benchmark popularity-debiasing methods change when such privacy level-based user personas are considered. One of the most important findings derived from the conducted comprehensive experimental studies is that disguising the actual preference data of individuals not only eliminates users’ concerns regarding their confidentiality but also makes the recommendations less biased towards item popularity at the same time. In other words, increasing the privacy level for users makes the collaborative recommenders more robust against the popularity bias problem. Thus, more sensitive users about their privacy can receive more qualified recommendations regarding beyond-accuracy aspects like diversity, novelty, catalogue coverage and fairness (i.e., equality in representing items), even though they bearably sacrifice from accuracy. Another important finding observed from our analysis is that as users’ privacy protection level improves, even if the performance of the considered popularity-debiasing methods enhances in featuring less-popular items in their recommendations, their effects on both beyond-accuracy and accuracy recommendation quality considerably diminish. Some of them even render impractical for individuals hypersensitive to their privacy.

When constructing privacy level-based user personas, we consider the randomized perturbation techniques as one of the most common approaches to provide user confidentiality. However, our analysis can be extended by considering alternative privacy-preserving methods, such as differential privacy and cryptographic techniques. Also, there is a need for novel popularity-debiasing methods that will not be affected by the data disguising procedure and thus show comparable performances for the scenario of users’ privacy is preserved.

Additional Information and Declarations

Competing Interests

Author Contributions

Data Availability

The authors declare there are no competing interests.

Mert Gulsoy conceived and designed the experiments, performed the experiments, analyzed the data, performed the computation work, prepared figures and/or tables, authored or reviewed drafts of the article, and approved the final draft.

Emre Yalcin conceived and designed the experiments, performed the experiments, analyzed the data, performed the computation work, prepared figures and/or tables, authored or reviewed drafts of the article, and approved the final draft.

Alper Bilge conceived and designed the experiments, analyzed the data, authored or reviewed drafts of the article, and approved the final draft.

The following information was supplied regarding data availability:

The codes are available at Zenodo: SiriusFoundation. (2023). SiriusFoundation/PopularityDebiasingWithPrivacy: Robustness of privacy-preserving collaborative recommenders against popularity bias problem (2023_03_v1.0). Zenodo. https://doi.org/10.5281/zenodo.7789914.

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
