# Peer review of "Robustness of privacy-preserving collaborative recommenders against popularity bias problem"

_PeerJ Computer Science, doi:10.7717/peerj-cs.1438_

## Round 0.1 · original submission · Major Revisions

This manuscript needs to introduce a privacy metric to evaluate the privacy protection of these ten recommendation algorithms. Please take all of the reviewers' points into consideration when revising your manuscript.

Reviewer 1 ·

Basic reporting

no comment

Experimental design

no comment

Validity of the findings

no comment

Additional comments

1) This study focuses on the recommendation system of imbalanced privacy-preserving collaborative filtering. This is an interesting topic for recommendation systems. However, some advanced work on imbalanced privacy-preserving federated learning needs to be introduced, such as, class-imbalance privacy-preserving federated learning for decentralized fault diagnosis with biometric authentication.

2) In section 2 (related work), as far as the privacy-preserving approaches for recommenders is concerned, some emerging technologies need to be introduced, such as block chain. The authors lack a review of related work in the last five years.

3) Privacy protection is at the core of this study. If possible, this manuscript needs to introduce a privacy metric to evaluate the privacy protection of these ten recommendation algorithms.

·

Basic reporting

The manuscript is well written, all around: problem framing, methods description, results and discussion, all are well and professionally explicated.

Experimental design

I can find no noteworthy methodological flaws. The datasets are not novel but as the authors point out, they are well known. Considering the research purpose, this is rather a strength than a weakness. Several algorithms and evaluation metrics are used, all relevant for the task.

Validity of the findings

Based on the observation that there are no noteworthy methodological flaws, the findings are valid.

Additional comments

Robustness of privacy-preserving collaborative recommenders against popularity bias problem

The study at hand tackles two important and worthwhile research topics within recommender algorithms: (a) privacy preservation and (b) popularity bias.

The manuscript is well written, all around: problem framing, methods description, results and discussion, all are well and professionally explicated.

I can find no noteworthy methodological flaws. The datasets are not novel but as the authors point out, they are well known. Considering the research purpose, this is rather a strength than a weakness. Several algorithms and evaluation metrics are used, all relevant for the task.

The literature review is adequate and well organized. The references are from reputable conferences, such as the ACM RecSys. References are, for the most part, recent.

In the following, I offer some suggestions for improving the manuscript:

(1) "Another primary limitation of CF recommenders is that it is challenging to collect confidential preference data of users since they usually avoid sharing their actual preferences about items due to their privacy concerns"
>> usually is too strong a claim, in my opinion. It is well known that users often rate items. The lack of ratings due to privacy concerns is not a major issue, in my opinion. There are other issues such as UI, conveniency and general laziness, but I'd argue against the assumption that "people usually avoid rating because of privacy". Of course, this depends on the context but for most consumer products and services, the problems are likely elsewhere. (People might not want to release demographic data but that's another thing.)

(2) "traditional CF recommenders are trained on the data consisting of unreliable ratings"
>> trained on data (no 'the') --- I'm not mentioning other grammar mistakes in the latter part of my review. Nevertheless, the general comment is: please have the manuscript professionally copyedited.

(3) "For this purpose, we specify four user categories according to their desired privacy-protection level. The first category, Utterly Unconcerned (UU), represents users who do not care about their privacy and require the highest possible level of accuracy in received recommendations. Note that UU also refers to users of traditional non-private CF recommenders. The second category, i.e., Barely Bothered (BB) corresponds to users who are not so worried if their privacy is violated, and they pick low-level privacy parameters in a request of good accuracy. The third category, Pretty Pragmatists (PP), represents users who do not compromise their privacy but are still interested in receiving decent recommendations. Therefore, they pick privacy parameters to balance accuracy and confidentiality. Finally, the fourth category, Fiery Fundamentalists (FF), resembles highly concerned people with their privacy, so they are only willing to share extremely perturbed preferences sacrificing accuracy completely."
>> loving this! Great way to make the comparison controlled but concrete. One comment: these could be called "personas" instead of "categories" (but this is just a question of concepts, feel free to ignore --- I'd call them personas since they describe user types!).

(4) "The next section presents a review of both privacy-preserving methods for CF algorithms and popularity bias concerned studies. The following section explains how we construct user categories according to varying privacy protection requirements. The next section explains the experimental setups, including utilized datasets, recommendation algorithms, and evaluation protocols, followed by experimentation methodology and considered popularity-debiasing methods. The following section"
>> the 'next section, following section' verbiage can simply be replaced by numbers: "Section 2 presents... Section 3 discusses..." (and so on)

I recommend accepting this work. Enjoyed reading it.

---

## Round 0.2 · accepted · Accept

According to the comments of reviewers and your reply, after comprehensive consideration, it is decided that accept.

Reviewer 1 ·

Basic reporting

The proposed scheme is presented in great detail, and the language is good.

Experimental design

The experimental design is logical and interesting.

Validity of the findings

The findings of this manuscript are valuable and compelling.

Additional comments

All my comments have been well-addressed. I recommend accepting this manuscript.

·

Basic reporting

Clear and unambiguous, professional English used throughout.
Literature references, sufficient field background/context provided.
Professional article structure, figures, tables.
Self-contained with relevant results.
Includes clear definitions of all terms.

Experimental design

Research question well defined, relevant & meaningful. It is stated how research fills an identified knowledge gap.
Rigorous investigation performed to a high technical & ethical standard.
Methods described with sufficient detail & information to replicate.

Validity of the findings

Datasets are well established.
Methods are well established.
Can see no validity issues.

Additional comments

The authors have addressed my comments. Recommending acceptance!